# Applications of CRISPR-Cas9 as an Advanced Genome Editing System in Life Sciences

**DOI:** 10.3390/biotech10030014

**Published:** 2021-07-06

**Authors:** Kamand Tavakoli, Alireza Pour-Aboughadareh, Farzad Kianersi, Peter Poczai, Alireza Etminan, Lia Shooshtari

**Affiliations:** 1Department of Medical Laboratory Science, Kermanshah Branch, Islamic Azad University, Kermanshah P.O. Box 6718997551, Iran; kamandtavakoli@yahoo.com; 2Seed and Plant Improvement Institute, Agricultural Research, Education and Extension Organization (AREEO), Karaj P.O. Box 3158854119, Iran; 3Department of Agronomy and Plant Breeding, Faculty of Agriculture, Bu-Ali Sina University, Hamedan P.O. Box 6517838695, Iran; f.kianersi@agr.basu.ac.ir; 4Botany Unit, Finnish Museum of Natural History, University of Helsinki, P.O. Box 7, FI-00014 Helsinki, Finland; 5Department of Plant Breeding and Biotechnology, Kermanshah Branch, Islamic Azad University, Kermanshah P.O. Box 6718997551, Iran; alietminan55@yahoo.com (A.E.); L_shooshtary@yahoo.com (L.S.)

**Keywords:** CRISPR system, Cas proteins, agriculture, animal science, human disease

## Abstract

Targeted nucleases are powerful genomic tools to precisely change the target genome of living cells, controlling functional genes with high exactness. The clustered regularly interspaced short palindromic repeats associated protein 9 (CRISPR-Cas9) genome editing system has been identified as one of the most useful biological tools in genetic engineering that is taken from adaptive immune strategies for bacteria. In recent years, this system has made significant progress and it has been widely used in genome editing to create gene knock-ins, knock-outs, and point mutations. This paper summarizes the application of this system in various biological sciences, including medicine, plant science, and animal breeding.

## 1. Introduction

Significant advancements have been made in biotechnology in recent years, and the branch of genetic engineering is advancing at an unprecedented pace, yielding numerous advantages. Genome editing technology has revolutionized genetic and biological re-search via the novel ability to precisely manipulate and modify the genomes of living organisms, and by accelerating the study of functional genomics. In recent years, different genome editing tools have been utilized to study simple and intricate genomes [1,2]. Genome editing technology emerged in the 1990s, and various methods have since been developed for targeted gene editing [3,4]. In general, three systems—each with their own advantages—have been widely used in cells and animals, including transcription activator-like effector nuclease (TALEN), zinc finger nuclease (ZFN), and clustered regularly interspaced short palindromic repeats (CRISPR) [3,5]. ZFNs are targetable DNA cleavage proteins with the ability to cut DNA sequences at any location [4]. TALENs comprise double-stranded breaks (DBs) in target sequences that trigger DNA damage response pathways that lead to repair [4]. Despite the widespread application of these tools during 2002–2012, some restrictions prevent their effectual use [4]. TALENs face problems with cloning large modules in series, reduced efficiency for screening positive targeted cells, and efficient ordered affiliate by ligase [3], whilst complexity and lack of specificity are big challenges for ZFNs [2,3,4]. A powerful genome editing system called CRISPR emerged in 1987, which became known as the greatest genetic tool of the century due to its outstanding advantages [6]. CRISPR-Cas9 technologies took the lead over other previous methods, such as TALENs and ZFNs, owing to its numerous advantages, like low cost, simplicity, high efficiency, and speed. Table 1 compares their differences [7]. In this work, a short summary of the CRISPR system and its application has been reviewed.

## 2. Origin, Development, and Mechanism of the CRISPR-Cas9 System

The CRISPR was first reported by Yushizumi Ishino [8], but its biological application was unknown at the time [3]. According to effector proteins, this system has been categorized into two main classes with six subtypes [2,9]. The type2 CRISPR-Cas9 system is the most widely used item in the field of genome editing with three main components: a CRISPR RNA (cRNA), an endonuclease named Cas9, and a transactivating crRNA (tracrRNA) [6]. This system consists of two components: (1) the Cas9 protein which can cleave the DNA and (2) the guide RNA that distinguishes the sequence of DNA to be rectified. To apply CRISPR-Cas9, sequences of the intended target genome are first identified. Then, the guide RNA is tailored to recognize a particular stretch of As, Ts, Gs, and Cs in the DNA. The guide RNA is affiliated to the DNA cutting enzyme Cas9, and then this complex is presented to the target cells. Cas9 locates the target letter and cuts the DNA at that point, allowing alteration of the existing genome by either modifying or adding to the sequence (Figure 1). As such, CRISPR-Cas9 functions as a cut and paste tool for DNA editing [10,11]. Using this technology, any genomic sequence identified by a short strand of guide RNA can be exactly modified [12]. This system targeted the human genome for the first time in 2013 [13,14,15]. To date, CRISPR-Cas9 has been commonly used to create gene editing in plants, animal, and human samples. This technique is widely used in various scientific fields, including medical science and therapeutics, as well as plant and animal sciences [16,17,18,19].

## 3. Ethical Issues in Genome Editing by CRISPR-Cas9 System

It is well understood in both scientific and related industries communities that CRISPR-Cas9 technology is one of the most important findings and inventions in the present century. However, due to its potential impact clinical applications and food safety several bioethical issues have arisen [20,21]. Indeed, the rapid rise of CRISPR-Cas9 over other technologies has led to new ethical issues in the various fields of bioscience [22,23]. Some bioethical issues related to the application of CRISPR-Cas9 are presented in Table 2.

## 4. Applications of CRISPER-Cas9 Technology

### 4.1. Human Science

Recently, CRISPR-Cas9 is being employed to study various genetic diseases, such as hemophilia [38], Duchenne muscular dystrophy [39,40,41,42,43], α1-antitrypsin deficiency [44,45], hearing loss [46,47], and hematopoietic [48,49,50] diseases. Figure 2 shows the application of this system in medical science. Hematologic diseases have previously been difficult to treat via genome manipulation. Studies in recent years demonstrated that CRISPR-Cas9 can correct genetic errors in hematopoietic stem cells that give rise to hematologic diseases [51,52,53,54,55]. These can then be applied to CRISPR-Cas9-based hematopoietic stem and progenitor cells (HSPCs) transplantation therapy. One of the most promising methods to treat hematopoietic diseases is editing the HBB mutation with CRISPR-Cas9, and it has been successfully carried out in patient-derived induced pluripotent stem cells (iPSCs) [56,57]. In a survey conducted by Park et al. [51], the CRISPR-Cas9 technology was used to rectify the *HBB* gene mutation in HSPCs. Final outcomes inferred a decrease in the amount of hemoglobin and sickle cells [17].

Since the CRISPR-Cas9 system originated from bacteria and acts as a bacterial immune system against invasive genetic agents, it has an inherent benefit in the treatment of bacterial and viral infection, and it is known as a new type of antiviral therapy against various incurable viral infections. Oncogenic viruses are highly related to carcinogenesis, including: Human papillomavirus (HPV), which could cause cervical cancer; Epstein Barr virus (EBV) that causes nasopharyngeal carcinoma; and hepatitis B virus that triggers liver cancer. CRISPR-Cas9 was first used to cleave HBV genomes [58]. It has previously reported that the production of the HBV core can decrease through the HBV expression vector [17]. Among human genetic diseases, cancer is the most common cause of death universally [6,17]. The tumor-suppressor genes have an important role in tumorigenesis. They can downregulate cancer progression by controlling cell proliferation and differentiation. CRISPR-Cas9 technology can target tumor-suppressor genes and restore them to interdict the tumorigenesis [59]. Table 3 indicates the extensive application of the CRISPR-Cas9 technique to knockout proto-oncogenes supporting the potential of this method in various tumors.

#### 4.1.1. Clinical Trials Using CRISPR-Cas9 Technology

The United States clinical trial database [60] contains useful information on experiments that have been used various genome editing tools such as ZFN, zinc finger, and CRISPR, Cas9, Cas12, and Cas13. Clinical trials using CRISPR-based therapies are still in their primary stages. It other words, if the treatments are safe and effective, they are likely to be several years away from legal authorities. New possibilities in precision medicine have arisen with the advent of CRISPR technology. Recently, many experiments are underway for several diseases such as several types of cancer, blood disorders, chronic infections, eye diseases, and protein folding disorders. All CRISPR-related trials have been performed to edit specific cells or tissues without affecting eggs or sperm; hence no DNA changes can be transferred on to future generations during these experiments [61,62]. The first application of CRISPR-based therapy refers to February 2019 in Germany. In this ex vivo trial, twelve patients were treated, and seven of them have been followed for at least three months. As a considerable result, they have observed that the patients had not needed blood transfusions even after treatment. Recently, Hirakawa et al. [62], Frangoul et al. [63], and Lu et al. [64] have provided a comprehensive information on application of CRISPR in the clinical trials. Herein, we summarized some clinical trials in a complete Table (Table 4).

#### 4.1.2. Clinical Trials of the Eye Based on CRISPR-Cas9

Clinical trials of in vivo genome editors have also begun. In all trials, tissues such as the eyes, cervix, and liver have been subjected to experiment. Cas9 was delivered to the eye using Adeno-associated virus vectors (AAV) as a therapy for congenital amaurosis type 10 LBA (LCA10). Because the most frequent LCA10 produces a mutation in the intron of the Centrosomal Protein 290 gene (CEP290) and provides a new binding site that changes the mRNA to form an early stop codon [65], it is a good candidate for the therapy of Cas9. Leber’s congenital amaurosis 10 therapy (LCA10) is reported as the first Cas9-related clinical trial (Table 4). This therapy employs two sgRNAs that, when combined, cause partial deletion of the intron or in-version of the partial intron, resulting in normal CEP290 protein production in the patient’s cells. Since the eye is an accessible tissue and subretinal injections in mice and mammals have provided stable gene editing, hence successful therapy may also be done in humans [65].

#### 4.1.3. Limitations of CRISPR-Cas-Based Gene Therapy

There are many projects related to CRISPR-Cas9 for disease research, and recent results and reports summarize the benefits of CRISPR-Cas9 [66,67]. Before employing CRISPR/Cas9-based gene treatments in humans, their safety and efficacy should be assessed and adjusted [68]. One of the main limitations of this technology is that not all mutation sites play a protospacer adjacent Motif (PAM) role, such that target detection depends entirely on this role. In addition, DNA damage response, delivery vehicle, editing at off-target genomic sites, and immunogenicity are other challenges of using CRISPR-Cas9 for gene therapy. The limitations of CRISPR-Cas9 technology are mainly related to extra-target DNA fracture formation, extra-target mutations, and PAM sequence dependence. Off-target effects are a serious problem in genome editing trials. The CRISP-Cas9 technology has a great chance of causing off-target alternations in human cells compared to other editing approaches [26,28,62,69]. Mutations outside the target location can induce gene dysfunction and, in rare cases, cell death. Choosing a suitable target locus possible during the bioinformatics analysis phase improves the effectiveness of the CRISPR-Cas9 system. Combining Cas9-D10A with sgRNA results in hydrolysis of one DNA strand at the target location [70].

### 4.2. Plant Science

Success in plant breeding depends on trait variability as well as overall genetic variation among plant populations. Recently, yield performance and several quality-related properties have been engineered by genome editing [71]. CRISPR-Cas9 has proved useful in crop improvement to increase disease resistance, improve yield, improve nutrition, and aid domestication. These advantageous genetic modifications and others will be discussed in this section.

#### 4.2.1. Plant Disease Resistance

Biotic stresses have usually been controlled by the spread of pathogen-resistant varieties and by applying agrochemicals. However, agrochemicals can cause contamination of the environment and negatively impact human health. Plant pathogens are also constantly evolving and can become unresponsive to these controls. Therefore, to create pathogen resistance in plants, resistance genes from wild species were introduced via breeding and established genetic transformation technologies involving large genomics regions [72]. Nonetheless, given the lack of specificity of the large genomic regions being introduced, other less-desirable traits may also be introduced using these approaches to create elite cultivars. Conversely, the CRISPR-Cas9 system provides a more accurate approach to genetic modification. Since first introduced to the field, it has proven to overcome several agricultural challenges including biotic stress resistance, fungal and bacterial disease resistance, and viral resistance [73]. For instance, CRISPR-Cas9 was employed to knockout the mitogen-activated protein kinase-5 (*OsMPK5*) gene to enhance disease resistance in rice [74]. Plant viruses invade a wide range of plants and affect the fertility of crops. As shown in Table 5, the CRISPR-Cas9 system provides a suitable context to develop resistance against both DNA- and RNA-based plant viruses [73].

As the latest genome editing tool, CRISPR-Cas9 has a key potential in developing bacterial resistance in plants. In several studies, the considerable potential for CRISPR-Cas9 technology has demonstrated to counteract crop bacterial diseases. (Table 6). For example, mutagenesis of the ethylene-responsive factor (ERF) transcription factor gene *OsERF922* using this technology enhanced resistance in rice cope with the blast disease. In another case, researchers modified the effector-connection element in the promoter of the *Lateral Organ Boundaries*-*1* gene (LOB) to confer resistance to citrus bacterial canker (CBC) caused by *Xanthomanas citri* subsp. in Duncan grapefruit [81].

#### 4.2.2. Yield of Crop Plants

Cereal crops play a key role in human life as food primarily supplying energy. Due to high demand for them, plant breeders have always sought ways to mass produce and generate products with high quality. With the CRISPR-Cas9 technology, it is possible to modify crop characteristics and improve their tolerance to adverse climate conditions and universal threats like drought, salinity, or frost [74]. For instance, to increase tolerance to salinity stress in rice, the *O. sativa* response regulator-22 gene (*osRR22*) was knocked-out to achieve approximately 65% mutation in T0 lines [74]. CRISPR-Cas9 was also described to hold potential for enhancing traits like drought or herbicide tolerance in maize [83]. As such, three sgRNAs were designed to target the *ZmTMS5* gene in maize and generate mutations in protoplasts, which resulted in edited plants showing bi-allelic modification [74]. Some research have shown the use of CRISPR-based genome editing in plants during the last decade, and a few studies have documented the use of genome editing to ameliorate biotic and abiotic stressors for crop development (Table 7).

#### 4.2.3. Genome Modification for Nutritional Improvement

As populations grow, the order for high value food crops increases [94]. The CRISPR-Cas9 system has made it possible to improve crop value and food quality through enhancing their nutritional status. For example, lycopene is a plant nutrient that is widely found in tomato with antioxidant properties and advantageous therapeutic traits. With the success gained in expanding the amount of lycopene content in tomato, it is expected that CRISPR-Cas9 technology may also play a key role in enhancing the micronutrient content of plants [73]. Other examples are presented in Table 8.

#### 4.2.4. Medicinal Plants

Medicinal plants may have significant effects in the treatment of various diseases, and they have always been used in traditional medicines. Terpenoids, coumarins, tannins, flavonoid, saponins, phenols, and cardiac glycosides are among the non-nutrient, bioactive, and physiologically active chemicals carried by plants. These phytocompounds are well-known for their health advantages. CRISPR-Cas9 could be used to modify targeted genes in herbal plants, survey the synthesis of efficacious com-pounds, select characteristics for increased yield, and advance research on biosynthetic pathways and regulatory mechanisms now that the genomes of some medicinal plants have been fully sequenced [75].

### 4.3. Animal Breeding

Due to population-growth driven increasing demand, the production of animal-derived food products, especially milk and meat, has increased worldwide. Providing these products is vital for the health and fitness of people [102]. Genome editing technology has made it possible to make precise changes in the animal genome to improve productivity and disease resistance [7]. One of the genes that was first genetically modified in farm animals was *myostatin*. Changes in this gene can drastically improve economic efficiency of meat production [7]. Farm animals that have been genetically engineered, thus far, include: pig, cattle, sheep, goat, and channel catfish [7]. Nonetheless, because of its global economic value, multiparous nature, and comparatively short generation time, pig is the most genetically modified livestock to date. CRISPR-Cas9 plays a key role in improvement of livestock as the most prominent gene-editing technology today [103,104] (Figure 3).

This method creates desired changes by either enhancing frequency of favorable alleles or by clearing deleterious alleles [7]. CRISPR-Cas9 has not only helped to increase animal products but also made many contributions to the field of biomedicine by producing transgenic and cloned animals [102]. Genome editing tools, like CRISPR-Cas9 and PiggyBac transposon, could be useful in immunology and vaccine development due to their low risk to human health, such as preventing the transition of viruses [102].

#### 4.3.1. Modification of Pigs for Xenotransplantation Research

The transplanting of living organs, tissues, and even living cells from one species to another is known as xenotransplantation or heterologous transplantation. The first serious xenotransplantation attempts were recorded in 1905, when slices of rabbit kidney were implanted into a boy with chronic renal disease [105]. Following that, in the first two decades of the twentieth century, many efforts to utilize organs from pigs, monkeys, and lambs were recorded [105]. Due to their considerable evolutionary distance from humans, pigs (*Sus scrofa domesticus*) are now the best prospects for organ donation among studied species. Indeed, this evolutionary distance reduces the probability of disease transmission across species [106]. Indeed, in addition to this issue, the short gestation period of pigs is the reason that they were considered as an animal model in clinical trials [107]. One of the most promising breakthroughs in pig-to-human xenotransplantation research is the silencing of porcine endogenous retroviruses (PERVs) and the insertion of the porcine *RSAD2* gene into the porcine *Rosa26* locus. Niu et al. [16] generated pigs in which all copies of *PERVs* were inactivated by CRISPR-Cas9 technology. Other examples regarding ap-plication of CRISPR technology can be seen in the literature, for example, Ryczek et al. [107], Niu et al. [108], Denner [109], and Hryhorowicz et al. [110].

#### 4.3.2. Application of CRISPR-Cas9 Technology in Insects

Most CRISPR-Cas genome editing research so far has relied on the *Streptococcus pyogenes* Cas9 nuclease, which can recognize the PAM sequence NGG [111]. Insects are the most numerous creatures found in nature. As a result, ongoing research efforts have added to the body of insect genomic information for species including *Drosophila melanogaster* Meigen, *Tribolium castaneum*, *Bombyx mori* L., *Apis mellifera* L., *Nasonia vitripennis* Walker, *Acyrthosiphon pisum*, and *Plutella xylostella* L. [112]. Insect uses of CRISPR-Cas9 are becoming more common, namely in *Drosophila melanogaster*, *Bombyx mori*, and *Aedes aegypti*. This method is not only becoming increasingly prominent in functional genomics research [113], but it is also being used as a tool for pest bug and vector-borne disease control [113,114]. Due to its capacity to have a wide variety of genetic tools, *Drosophila* has been regarded as one of the greatest insect models for the study of eukaryotic biology, including human development and illness in insect research. Gratz et al. [115] started the *Drosophila* CRISPR-Cas9 journey by deleting a 4.6 kb segment of the yellow gene using two gRNAs targeting the 5′ and 3′ ends of the source, respectively, in 2013. CRISPR-Cas9 is also being tested in other insect species, according to researchers (Table 9).

## 5. Conclusions

The CRISPR-Cas9 technology is a new and purposeful method in genome editing that has recently surpassed other methods due to its outstanding advantages. So that, it has significantly contributed to all fields of life science, including medicine, plant breeding, and animal breeding, also expanding researchers’ understanding of the basis of gene diversity and gene editing. Given its current central role in this landscape, ongoing research will likely focus on improving this technology further to enhance its specificity and efficiency.

## Figures and Tables

**Figure 1 biotech-10-00014-f001:**
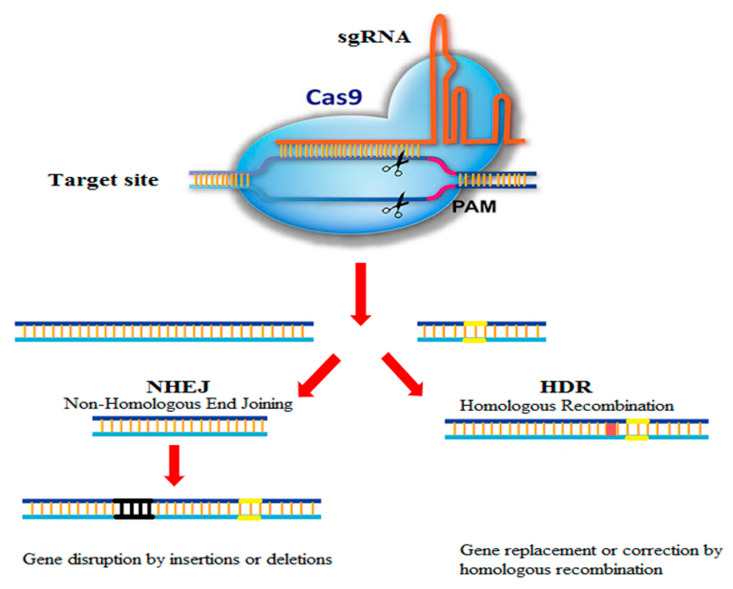
Schematic of CRISPR-Cas9 genome editing system.

**Figure 2 biotech-10-00014-f002:**
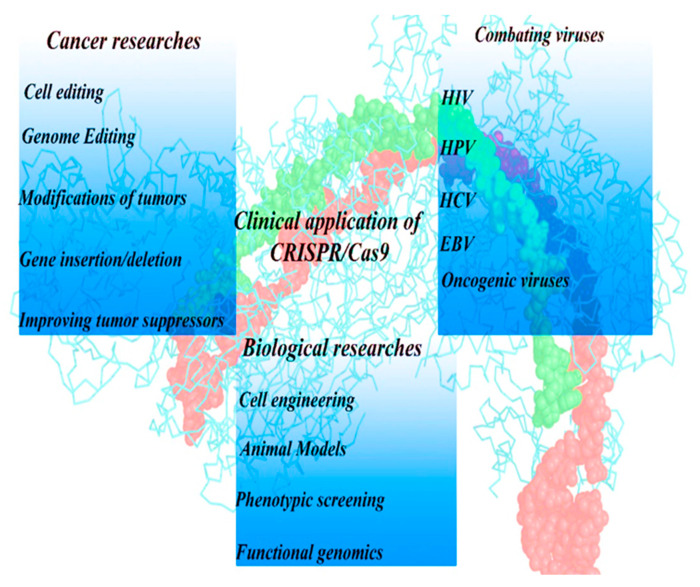
Exploitation of the CRISPR-Cas9 system in the field of medicine, including infectious diseases, tumors, and genetic diseases.

**Figure 3 biotech-10-00014-f003:**
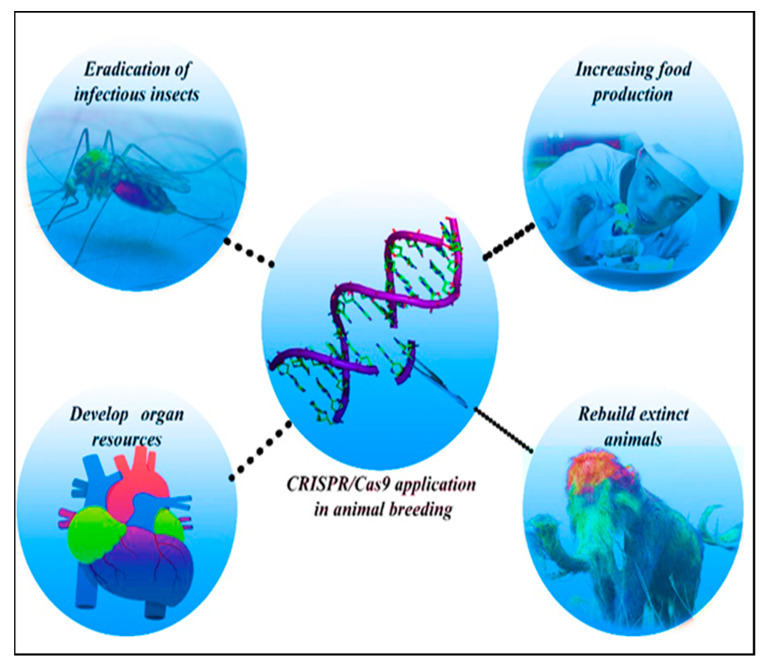
Exploitation of CRISPR-Cas9 in animal breeding and animal models.

**Table 1 biotech-10-00014-t001:** Main differences between three genome editing techniques.

Feature	CRISPR-Cas	TALEN	ZFN
Cost	Low	High	Low
Ease of design	Simple	A little complex	Moderate
Specificity	High	Intermediate	Low
Pros	Modifies multiple sites in tandem	Highly effective and specific	Highly effective and specific
Cons	PAM motif required next to target sequence	Time consuming	Time consuming
Multiplex genome editing	High-yield multiplexing	Few models	Few models

**Table 2 biotech-10-00014-t002:** Some of the bioethical issues and possible risks intended for the application of the CRISPR-Cas9 system.

Organism	Risks	Bioethical Issues	References
Bacteria	Gene mutations/ Gene drifts	Disruption of ecological balance	[20,24,25]
Plants	Gene mutations/ Gene drifts	Disruption of ecological balance	[20,26]
Animals/ chimeric animals	Gene mutations	Disruption of ecological balance	[24,27,28,29,30,31,32]
Humans	Gene mutations Side effects Cost Genetic mosaicism	Eugenics Informed consent Enhancement Accessibility Patenting Safety Incomplete or over legislation	[21,24,26,33,34,35,36,37]

**Table 3 biotech-10-00014-t003:** The use of CRISPR-Cas9 system in the treatment of different types of cancer [59].

Type	Oncogene	Tumor Suppressor Gene	Drug-Resistance Gene
Breast	*SHCBP1, MIEN1, miR-27b, mi23b, HER2 exons*	*CPEB2, ETS1, BRCA1*	*HER2, EGFR, ER*
Prostate	*PCAR19, CECAM21, SENP1, miR-302/367, miR-1205*	*PGC1a, DEPTOR, p53, RB1*	*BLS-211, NANOGP8, NANOG1*
Lung	*IGFIR, ERBB2, RSF1, FOS, MCM4*	*MFN2, MiR-1205, GOT1, TP53*	*NRF2, MiR-1205, ER300, RSF1*
Liver	*Plxnb1, NCAPG, CDK7, IncBRM, Nf1*	*BAP1, HELLS, Tp53, Traf3*	*NF1, MED12, ERK2*
Colorectal	*KRAS, HPV16, Fut4, NRAS*	*LIMCH, PTEN, SOX15*	*miR-139-5P, ZEB1*
CRISPR-Cas9 function	Knockout	Activate	Promote drug sensitivity

**Table 4 biotech-10-00014-t004:** Several examples for clinical trials with genome editor of CRISPR-Cas9.

Target Gene and Effect	Disease	Intervention
Cas9-mediated creation of CD19 and CD20	Leukemia	CAR T cells to CD19 and CD20 or CD19 and CD22
CCR5 knockout	HIV	Modified CD34+ hematopoietic stem cells
CD7 knockout in CD7 CAR T cells	T-cell malignancies	CAR T cells to CD7 and knockout of native CD7 to prevent self-targeting
Correction of the hemoglobulin subunit β globulin gene	β-thalassemia	Ex vivo modified hematopoietic stem cells
Creation of a CD19-directed T cell	Refractory B-cell malignancies	CD19-directed T-cell immunotherapy
Cytokine-induced SH2 protein (CISH) knockout	Metastatic gastrointestinal epithelial cancer	Modified tumor-infiltrating lymphocytes
disruption of HPK1	Refractory B cell malignancies	CD19-CAR modified T cells with CAR delivered by lentivirus and Cas9 knockout of HPK1
Disruption of the erythroid enhancer to BCL11A gene	β-thalassemia	Ex vivo modified hematopoietic stem cells
	Sickle cell anemia	
	β-thalassemia and severe sickle cell anemia	Ex vivo- modified hematopoietic stem cells, 15-year follow-up study
E6 and E7 oncogene of HPV16 and HPV18 deletion	HPV-related malignancy	Plasmid in a gel containing a polymer to facilitate delivery
Programmed cell death protein 1 (PD-1) knockout	Mesothelin positive solid tumors	CAR T cells to mesothelin with added PD-1 and TCR knockout
	Hormone refractory prostate cancer	Modified T cells
	Esophageal cancer	
	Metastatic non-small cell lung cancer	
	Stage IV bladder cancer	
	Metastatic renal cell carcinoma	
	EBV-positive, advanced stage malignancies	Modified T cells selected for those targeting EBV positive cells
	Mesothelin positive solid tumors	CAR T cells to mesothelin with PD-1 knockout
Removal of alternative splice site in CEP290	Leber congenital amaurosis 10	ZFN-mediated removal of intronic alternative splice site in retinal cells
TCRα, TCRβ, PD-1 knockout	Various malignancies	Modified T cells with Cas9-mediated deletions and lentiviral transduction of NY-ESO-1 targeted TCR
βTCRα, TCRβ, β-2 microglobin (B2M) knockout	B-cell leukemia	CD19-CAR modified T cells with CAR delivered by lentivirus and Cas9 knockout B2M and TCR to create universal T cells

**Table 5 biotech-10-00014-t005:** Proven viral resistance in plants introduced via CRISPR-Cas9 against DNA and RNA viruses [73].

Virus	Type of Nucleic Acid	Involved Protein	Plant under Attack	References
Beet severe curly top virus	DNA	Cas9	Capsicum	[75]
Bean yellow dwarf virus	DNA	Cas9	Oat	[76]
Turnip mosaic virus	RNA	Cas13	Cruciferous plants, Chinese cabbage, turnip, mustard, radish	[77]
Tomato yellow leaf curl virus	DNA	Cas9	Invading a number of seeds, including tomato	[78,79]
Yellowing virus	RNA	Cas13	Cucumber	[80]
Zucchini yellow mosaic virus	RNA	Cas13	Cucumber	[81]
Papaya ring spot mosaic virus	RNA	Cas13	Cucumber	[82]

**Table 6 biotech-10-00014-t006:** Three examples of exploitation of CRISPR-Cas9 to counteract crop bacterial disease.

Plant	Targeted Area in Gene	Disease	References
Rice	Mutagenesis of the ERF Transcription Factor Gene *OsERF922*	Blast	[72]
Duncan grapefruit	Effector-binding element in the promoter of the Lateral Organ Boundaries 1 gene	Citrus bacterial canker (CBC)	[81]
Wanjinchen oranges	(*CsLOB1G* and *CsLOB1−*) alleles	Citrus bacterial canker (CBC)	[82]

**Table 7 biotech-10-00014-t007:** Several examples of the application of CRISPR-Cas9 technology in plants against environmental stresses.

Crop	Method	Target Gene	Stress/Trait	References
*A. thaliana/* *N. benthamiana*	NHEJ	dsDNA of virus (A7, B7, and C3 regions)	Beet severe curly top virus resistance	[75]
*N. benthamiana* Bean	NHEJ	BeYDV	Yellow dwarf virus (BeYDV) resistance	[76]
*N. benthamiana*	NHEJ	ORFs and the IR sequence sDNA of virus	Tomato yellow leaf curl virus (TYLCV) and Merremia mosaic virus (MeMV)	[77]
Rice	NHEJ	*OsERF922* (ethylene responsive factor)	Blast Resistance	[78]
Cucumber	NHEJ	*eIF4E* (eukaryotic translation initiation factor 4E)	Cucumber vein yellowing virus (CVYV), Zucchini yellow mosaic virus (ZYMV), and (PRSV-W)	[80]
*A. thaliana*	NHEJ	*eIF(iso)4E*	Turnip mosaic virus (TuMV) resistance	[84]
Rice (IR24)	NHEJ	*OsSWEET13*	Bacterial blight disease resistance	[85]
Bread wheat	NHEJ	*TaMLO-A1*, *TaMLO-B1,* and *TaMLOD1*	Powdery mildew resistance	[86]
Maize	HDR	*ARGOS8*	Increased grain yield under drought stress	[87]
Tomato	NHEJ	*SlMAPK3*	Drought tolerance	[88]
*A. thaliana*	HDR	*MIR169a*	Drought tolerance	[89]
*A. thaliana*	NHEJ	*OST2* (OPEN STOMATA 2) (AHA1)	Increased stomatal closure in response to abscisic acid (ABA),	[90]
Rice	HDR/ NHEJ	*OsPDS*, *OsMPK2*, *OsBADH2*	Involved in various abiotic stress tolerance	[91]
Rice	NHEJ	*OsMPK5*	Various abiotic stress tolerance and disease resistance	[92]
Rice	NHEJ/ HDR	*OsMPK2, OsDEP1*	Yield under stress	[93]

**Table 8 biotech-10-00014-t008:** Application of CRISPR-Cas9 technology in plants for nutritional traits.

Crop	Method	Target Gene	Stress/Trait	References
Rice	NHEJ	2.5604 gRNA for 12,802 genes	Creating genome wide mutant library	[95]
Maize	NHEJ	*ZmIPK1A ZmIPK* and *ZmMRP4*	Phytic acid synthesis	[96]
Wheat	HDR	*TaVIT2*	Fe content	[97]
Soybean	NHEJ	*GmPDS11* and *GmPDS18*	Carotenoid biosynthesis	[98]
Tomato	NHEJ	*Rin*	Fruit ripening	[99]
Potato	HDR	*ALS1*	Herbicide resistance	[100]
Cassava	NHEJ	*MePDS*	Carotenoid biosynthesis	[101]
Rice	NHEJ	2.5604 gRNA for 12,802 genes	Creating genome wide mutant library	[95]
Maize	NHEJ	*ZmIPK1A ZmIPK* and *ZmMRP4*	Phytic acid synthesis	[96]
Wheat	HDR	*TaVIT2*	Fe content	[97]
Soybean	NHEJ	*GmPDS11* and *GmPDS18*	Carotenoid biosynthesis	[98]
Tomato	NHEJ	*Rin*	Fruit ripening	[99]
Potato	HDR	*ALS1*	Herbicide resistance	[100]
Cassava	NHEJ	*MePDS*	Carotenoid biosynthesis	[101]

**Table 9 biotech-10-00014-t009:** Applications of CRISPR-Cas9 in insects.

Species	Targeted Genes	Strategy	Germline Transmission Rate (%)	G1 Mutation Rate (%)	References
*Drosophila* spec.	yellow, whit	mRNA INJ	0–79	0–34.5	[116]
	CG4221, CG5961, Chameau	mRNA INJ with donor	8.1–26.7	2.7–10.4	[117]
	yellow	DNA INJ with donor	5.9–20.7	0.25–1.37	[115]
	yellow	Rapid INJ with donor	8–53	15	[118]
*Bombyx mori*	BmBLOS2	mRNA INJ	95.5	35.6	[119]
	th, re, fl, yellow-e, kynu, ebony	DNA INJ	5.7–18.9	ND	[120]
*Aedes aegypti*	ECFP	mRNA INJ + DNA INJ	0	5.5	[121]
*Daphnia magna*	eyeless	mRNA INJ	18–47	8.2	[122]
*Tribolium castaneum*	eGFP1	mRNA INJ + DNA INJ with donor	55–80	71–100	[123]
*Papilio xuthus, P. machaon*	abdominal-B, ebony, frizzled	mRNA INJ	18.33–90.85	ND	[124]

NIJ and ND indicate injection and not determined, respectively.

## Data Availability

Not applicable.

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
