# Peer review of "Applications of CRISPR-Cas9 as an Advanced Genome Editing System in Life Sciences"

_biotech, 2021, doi:10.3390/biotech10030014_

Round 1

Reviewer 1 Report

In this manuscript titled, "Applications of CRISPR-Cas9 as an advanced genome editing 2 system in life science", Kamand Tavakoli et al., authors summarized the application of the CRISPR-Cas9 system in various biological sciences, including medicine, plant science, and animal breeding. This manuscript is written clearly, however, the manuscript appears preliminary.

  1. There already have lots of reviews about the applications of CRISPR-Cas9, what are the new points in this manuscript compare to other reviews?
  2. Are there any applications of CRISPR-Cas9 in COVID-19?

Author Response

Dear Reviewer,

Thank you for forwarding your comments and suggestions. We are pleased to know that our manuscript was rated as potentially acceptable for publication in this journal. In the revised text, all changes have been highlighted in red However, we responded to your questions as below. 

1# There already have lots of reviews about the applications of CRISPR-Cas9, what are the new points in this manuscript compare to other reviews?

Response: As you mentioned, to date, there are numerous reports about CRISPR/Cas 9 as well as application of it in different organisms such as human, plants, and etc. However, the current work is one of the manuscripts that reviewed, collected, and integrated the application information and efficiency of CRISPR/Cas9 in humans, plants, and animals, for future studies.

2# Are there any applications of CRISPR-Cas9 in COVID-19?

Response: Thank you for your point. As you know there is important reposts and manuscripts about the application of CRISPR/Cas technology, CRISPR/Cas 12 and 13, to study and cure the SARS-CoV-2 (COVID-19) (for example Nguyen et al. [2020]; Abbott et al. [2020]; Broughton et al. [2020]), but there is not any report and manuscript about application of CRISPR/Cas 9 method to treatment SARS-CoV-2 (COVID-19) diseases as exactly. Hence, due to this point that there have been reports on the induction of mutations in target sites by CRISPR-based antivirals using Cas9 while no crRNA target-site mutation was observed following Cas13 treatment. Thus, these results demonstrate the potential of the Cas13 enzyme as an effective antiviral for only the management of SARS-CoV-2 infection.

Reviewer 2 Report

Interesting, however very short, review article.

As major points to be further elucidated:

  • add a paragraph highlighting ethical considerations on the use of CRISPR technology in humans and in plants;
  • limitations should also improved;
  • if any, describe current clinical trials using this technology.

Author Response

Dear Reviewer,

Thank you for your forwarding comments and suggestions. We are pleased to know that our manuscript was rated as potentially acceptable for publication in this journal. We have revised the manuscript by addressing all your comments and suggestions through the text and highlighted them in green.

Reviewer 3 Report

In this manuscript, the authors describe some applications of CRISPR-cas9 in the edition of different organisms: human, plants, and animals. The manuscript needs to be improved. I detail some considerations below.

Major issues:

You should include in this revisión the weakness not only the strength of CRISPR technology. For example immunity response, PAM sequence dependence etc. Moreover, you should include data about reagents delivery. The authors focused on cancer but the application of CRISPR is more advanced in other pathologies, that should be taken into account, for example, eye diseases. You should also include insect and microorganisms editing by CRISPR because they are very important in biotechnology.

In addition, some references should be reconsidered because I think they are not the most appropriate.

Other considerations:

1-Line50 the reference 3 must be changed by  Doudna JA and Charpentier E. Science 2014, for example. They obtained the Nobel Prize in Chemistry 220 for the development of a method for genome editing. On the other hand, the first evidence of CRISPR biological function was. Barrangou, R., et al., CRISPR provides acquired resistance against viruses in prokaryotes. Science, 2007.

2-Line61. You should change the reference.

3-Line 82 You should include some references. You should also remark that all of the applications indicated are only experimental and not clinical applications.

4-Line 85. You should include some references.

5-Line 92 You should revise the reference.

6- Authors have included tables describing samples of two applications: to introduce viral resistance and to counteract bacterial diseases, although other important applications should be remarked like natural environmental stress resistance, conservation parameters, etc, that should be included in other additional tables

7- Line139 You should include reference [30]

8- Modification of organisms (pigs)  for xenotransplants is an important CRISPR application that should be included in the manuscript.

9-The phrase “CRISPR-Cas9  technologies took the leading role over other previous methods such as TALENs and ZFNs, owing to numerous advantages, like low cost, simplicity, high efficiency, and speed. Table 4 compares their differences [34]” should move to another part of the manuscript.

Author Response

Dear Reviewer,

Thank you for your forwarding comments and suggestions. We are pleased to know that our manuscript was rated as potentially acceptable for publication in this journal. We have revised the manuscript by addressing all your comments and suggestions through the text and highlighted them in pink.

Round 2

Reviewer 2 Report

Authors made a good job in revising their manuscript that is acceptable for publication in my opinion.

Just check if it is correct to cite the name of the SCD patient (see page 6, line 222 and ref#64).

Author Response

line 195, the sentence " The CRISPR-Cas9 system has a higher risk of developing off-target mutations in human cells compared to the zinc finger nuclease (ZFN) and TALE nuclease" maybe should be changed taking into account there are many papers describing that there is not off-target using CRISPR.

Response: Thank you for your positive opinion and your suggestion. With respect to the referee, a published articles on efficiency of CRISPR system are various that each of them evaluated different aspects of advantages and disadvantages of the CRISPR rather than the previous edition systems. CRISPR system has been used successfully to edit eukaryotic genomes as well as it has substantially accelerated the understanding of the functional organization of the genome at the systems level. However, many of study and research has been reported about the limitation and problems of CRISPR method and off-target effects (Hsu et al., 2013; Cho et al., 2014; Zhang et al., 2015; Rodriguez, 2016;  Shinwari et al., 2017; Aryal et al., 2018, Wienert et al., 2019; Newton et al., 2019; Hirakawa et al., 2020). On the other hand, off-target mutations observed at frequencies greater than the intended mutation, which may cause genomic instability and disrupt the functionality of otherwise normal genes, is still one major concern about off-target effects when applying CRISPR/Cas9 system to biomedical and clinical application, since they may have deleterious effects on humans and the environment.

Reviewer 3 Report

The authors have added my suggestions and the paper is now ready to be published.

Only one minor consideration:

Line 195, the sentence: “The CRISPR-Cas9 system has a higher risk of developing off-target mutations 195 in human cells compared to the zinc finger nuclease (ZFN) and TALE nuclease” maybe should be changed taking into account there are many papers are describing that there are not off-target using CRISPR

Author Response

line 195, the sentence " The CRISPR-Cas9 system has a higher risk of developing off-target mutations in human cells compared to the zinc finger nuclease (ZFN) and TALE nuclease" maybe should be changed taking into account there are many papers describing that there is not off-target using CRISPR.

Response: Thank you for your positive opinion and your suggestion. With respect to the referee, a published articles on the efficiency of the CRISPR systems are various that each of them evaluated different aspects of advantages and disadvantages of the CRISPR rather than the previous edition systems. The CRISPR system has been used successfully to edit eukaryotic genomes as well as it has substantially accelerated the understanding of the functional organization of the genome at the systems level. However, many of study and research has been reported about the limitation and problems of the CRISPR method and off-target effects (Hsu et al., 2013; Cho et al., 2014; Zhang et al., 2015; Rodriguez, 2016;  Shinwari et al., 2017; Aryal et al., 2018, Wienert et al., 2019; Newton et al., 2019; Hirakawa et al., 2020). On the other hand, off-target mutations observed at frequencies greater than the intended mutation, which may cause genomic instability and disrupt the functionality of otherwise normal genes, is still one major concern about off-target effects when applying CRISPR/Cas9 system to biomedical and clinical application, since they may have deleterious effects on humans and the environment.

Just check if it is correct to cite the name of the SCD patient (see page 6, line 222 and ref 64).

Response: Thank you for your guide and your positive opinion. The name of SCD is correct to cite in the text and the reference. SCD is the abbreviation of the "sickle cell disease" word, in the manuscript.
